# PSMA PET-CT in the Diagnosis and Staging of Prostate Cancer

**DOI:** 10.3390/diagnostics12112594

**Published:** 2022-10-26

**Authors:** Alexander D. Combes, Catalina A. Palma, Ross Calopedos, Lingfeng Wen, Henry Woo, Michael Fulham, Scott Leslie

**Affiliations:** 1Department of Urology, Royal Prince Alfred Hospital, Sydney, NSW 2050, Australia; 2Department of Molecular Imaging, Royal Prince Alfred Hospital, Sydney, NSW 2050, Australia; 3Faculty of Engineering and Computer Science, University of Sydney, Sydney, NSW 2006, Australia; 4Faculty of Medicine, University of Sydney, Sydney, NSW 2006, Australia; 5Department of Urology, Chris O’Brien Lifehouse, Sydney, NSW 2050, Australia; 6RPA Institute of Academic Surgery, Sydney, NSW 2050, Australia

**Keywords:** prostate cancer, imaging, PSMA PET

## Abstract

Prostate cancer is the most common cancer and the second leading cause of cancer death in men. The imaging assessment and treatment of prostate cancer has vastly improved over the past decade. The introduction of PSMA PET-CT has improved the detection of loco-regional and metastatic disease. PSMA PET-CT also has a role in the primary diagnosis and staging, in detecting biochemical recurrence after curative treatment and in metastasis-directed therapy. In this paper we review the role of PSMA PET-CT in prostate cancer.

## 1. Introduction

Since the advent of PSA testing, prostate cancer management has been fast evolving and heavily debated. This is in part due to the high prevalence but protracted course of the disease, coupled with our understanding of tumour biology—an enigma that is inherently limited by technology used to assess it. PSMA PET-CT is an example of technology that has influenced our practice. It has resulted in a divergence from traditional algorithms, opened the realm of theranostics and highlighted as many uncertainties as it has improvements.

By all measures, PCa is the most common cancer in men and one of the most common cause of cancer deaths amongst men [1,2]. Routine PSA testing was met with resistance by opponents touting PCa is an indolent disease that men usually die with, rather than from [3,4]. This is not a surprising sentiment considering conflicting mortality benefits from screening and early treatment published in international literature [5,6,7]. Unfortunately, the resultant drops in PSA testing also correlated with increased prostate cancer mortality [8]. Furthermore, societal impact of men living with metastatic disease is unequivocal. Approximately, two million men each year are diagnosed with prostate cancer with 10 million men living with the disease and 700,000 of these living with metastatic disease [9]. Analysis of conflicting organisational policies and management algorithms requires historical examination of clinical practices.

Traditionally, prostate cancer risk stratification relied predominantly on PSA level & kinetics, digital rectal examination (DRE), and Gleason score obtained from non-targeted ‘sextant’ template biopsy [10]. PSA screening led to early treatment of prostate cancer, leading to a decline in mortality, but the limited specificity of PSA and DRE likely resulted in overdiagnosis and overtreatment [10]. Furthermore, the classic TRUS sextant biopsy probably missed a proportion of clinically significant cancers and detected cancers that we now know are unlikely to warrant immediate treatment [11]. In addition to a high overall error rate in accurate diagnosis of prostate cancer, staging of PCa was also limited by the poor accuracy of CT and bone scan. Prior to development of MRI and PSMA PET-CT, a variety of nomograms were then developed to help risk stratify patients diagnosed with PCa. For example, the NCCN guidelines use PSA, Gleason score, and clinical stage to stratify PCa into ‘very low’, ‘low’, ‘intermediate’, ‘high’, and ‘very high’ risk categories, which guide clinical management before and after definitive therapy [12]. Within the limits of our understanding then, changing nomenclatures and evolving nomograms sought to better guide our practice.

With the same momentum, the focus partly diverged from the traditional categories and subclassifications, which are founded on light-microscopy glandular architecture and serum PSA levels [13]. The international community embarked on a deeper molecular and genetic understanding of PCa, the results of which are difficult to transplant into current practice [14]. Notably, a variety of serum, urine, and biopsy biomarkers have also been approved to more accurately identify men with high-grade PCa [15]. Similarly, imaging technology evolved to investigate more than cross-sectional architecture alone. The advent of multi-parametric MRI enriched the risk stratification algorithm in this way. Improved anatomical detail coupled with sequences used to differentiate tissue characteristics of internal structures has allowed targeted sampling that has been shown to yield more clinically significant disease [16,17]. Some have even proposed the combination of Pi-RADS score and mRNA urine test to improve PCa detection further [18].

Combining progress in molecular oncology and imaging technology, PSMA PET-CT represents the next step in our pursuit to better treat men with PCa. It has similarly superseded traditional staging techniques of CT and bone scan and has been shown to enhance local staging of MRI. By coupling bio-functional and anatomical information, we can more accurately detail the extent of a patient’s disease. The broad application of its utility in modern PCa management will be outlined in the article. For reference PSMA PET-CT will refer to ^68^GA- PSMA PET-CT in this article unless specified otherwise

## 2. Positron Emission Tomography-Computed Tomography (PET-CT)

PET began in the 1980s and employed the tracer-kinetic assay method and tomographic image reconstruction to provide images of ‘function’. The first PET scanners were built ‘in-house’ in various research institutions in the northern hemisphere. PET-CT was introduced as a clinical tool in late 2000 and replaced PET-only scanners. Positrons are positive electrons, which are emitted from the nucleus of positron emitters (^18^F, ^15^O, ^11^C, ^13^N), have short half-lives and are usually produced by medical cyclotrons. The positron emitters are ‘tagged’ or attached to compounds using complex synthetic modules to produce a PET radiopharmaceutical. The PET radiopharmaceutical then, usually after intravenous injection in humans in trace amounts (hence, the term ‘PET tracer’), participates but does not perturb, a biochemical process of interest such as glucose metabolism or receptor uptake. Positron emitters decay to a stable state by the emission of a positron, which when in tissue collides with an innocent bystander electron; this collision results in the annihilation of the electrons and the generation of two 511 keV gamma rays (photons) that are then detected by crystal detectors in the PET tomograph [19]. In prostate cancer, the most commonly used PET radiopharmaceutical is ^68^Ga-PSMA; ^68^Ga has a half-life of 68 min. ^68^Ga is usually produced from a ^68^Ga-^68^Ga generator with a few exceptions—our institution and a few other sites in the world produce ^68^Ga in a cyclotron [20]. In clinical practice, PET-CT imaging is usually undertaken at a set time, after the intravenous injection of the PET radiopharmaceutical, referred to as the ‘uptake time’. The uptake time for ^68^Ga-PSMA in our institution is 50 min. The uptake of the PET radiopharmaceutical in the tissue can be quantified by kinetic analysis but this requires continues scanning after the injection of the PET radiopharmaceutical and this consumes valuable time on the scanner. So, for the clinic, a semi-quantitative measurement of the uptake, the standardized uptake value (SUV), that relates uptake to the injected activity of the PET radiopharmaceutical, the patient’s weight and to time, is used.

Since 2000 there have been progressive improvements in the PET and CT technologies—smaller more efficient crystals, digital photomultiplier tubes in PET; faster CT scanners with 64- and 128-images slices; better reconstruction techniques, improvements in ‘time-of-flight’ timing and continuous bed motion. The most recent advance has been the development of long field of view PET-CT scanners or ‘total body’ scanners. Siemens Healthineers have introduced the Biograph Vision Quadra (the ‘Quadra’) a scanner with a 106 cm *z*-axis field-of-view (FOV) and United Imaging, the Explorer with a 200 cm FOV. Both scanners have markedly improved sensitivity enabling faster scans, lower injected doses of PET radiopharmaceuticals and superior image quality. There are less than a dozen such scanners in use at the present time. The majority of the images used in this paper are from our Biograph Vision Quadra which was the 2nd such device installed and it went ‘live’ in May 2021. The sensitivity of the Quadra is 16× that of conventional PET-CT scanners and it allows the simultaneous acquisition of data throughout the *z*-axis extent of the scanner.

## 3. PSMA PET-CT

PSMA is a surface receptor antigen expressed in prostate tissue and tumour-associated neovasculature [21]. It is a glutamate carboxypeptidase type II non-secreted transmembrane protein comprising 750 amino acids. Antibodies that bind to the extracellular domain of PSMA were developed. The first was humanised IgG monoclonal antibody ‘J591′, followed by the DKFZ- PSMA-617 ligand and the peptide-linker unit DOTAGA-(I-y)fk (Sub-KuE), termed PSMA-I & T [22] and these were then radio-labelled. Most published studies report ^68-^Ga labelled compound but recently ^18^F-labelled PSMA have been used [23]. Of relevance to the interpretation of PSMA PET-CT scans, PSMA expression is found in normal prostatic tissue albeit to a mild degree, in the salivary and lacrimal glands, nasal space and larynx, liver, spleen, bowel, the kidneys and the sympathetic ganglia (Figure 1) (20). PSMA uptake is also found in other tumors including glioblastoma, thyroid, breast, lung, colon and renal carcinomas. PSMA uptake is also seen in benign tumors—haemangiomas, thyroid and adrenal adenomas, schwannomas and desmoid tumors—and also in reactive/inflammatory conditions and Paget’s disease [21]. PSMA is overexpressed in almost all PCa by around 100–1000 times the normal level, however the exact reason for this remains unclear [24,25]. Current hypothesis suggests that functionally PSMA has a role in folate metabolism, with the extra-cellular unit hydrolyzing glutamated folates released by PCa cells, which are then utilised to enhance proliferation in PCa cells [26]. It’s utility in the diagnosis and staging of prostate cancer has since become an extremely advantageous imaging modality furthering international urological practice.

## 4. Diagnosis

Multiparametric MRI (mpMRI) has improved our ability to select who will benefit from a prostate biopsy and how to get best yield from that biopsy. The technique has reported sensitivities of up to 91% for grade group ≥2 PCa and 95% for grade group ≥3 [27]. A meta-analysis of over 15 studies involving men with suspected PCa, showed the average positive predictive value (PPV) in mpMRI alone for ISUP ≥ 2 with PIRADS score of 3,4 and 5 were 16%, 59% and 85%, respectively [28]. Several prospective trials including the PRECISION trial also found that MRI-targeted biopsy had far superior levels of detecting PCa with ISUP ≥ 2 compared to the systematic biopsy approach or MRI alone [29,30].

Like any new technology, the application of mpMRI is not perfect. Ambiguity still exists about whether biopsy is appropriate for PI-RADS 3 lesions. Specifically, PI-RADS v2.1 includes modifications in the assessment of lesions in the central zone and the anterior fibromuscular stroma, evaluation of the transition zone, and revision of the criteria for characterizing lesions as 2 or 3 on DWI [17]. Despite this, accurate interpretation of very large hyperplastic transitional zones remains difficult. Prostatitis is a common and false-positive/false-negative rates as high as 14% for ISUP ≥3 PCa have been reported, indicating that significant prostate cancer can be missed for 1 in 7 patients [31]. Additionally, cases of MRI-invisible lesions have been reported [17]. PET imaging is being used to clarify difficult cases and augment this space.

Previous nuclear medicine studies using FDG-PET lacked the sensitivity and specificity to successfully diagnose clinically significant PCa. A large retrospective study investigating 47,109 men who underwent FDG-PET over a 10-year period found 1335 patients with incidental FDG uptake in the prostate. Only 1 patient with a normal PSA (<2.5 ng/mL) and just 40 patients with elevated PSA out of 93 had biopsy confirmed PCa [32]. It was not until the advent of antibodies to PSMA and thereafter the introduction of PSMA PET-CT, have we been able to identify localised prostate cancer with sufficient accuracy and avoid excessive and unnecessary prostatic biopsies.

A recent meta-analysis also showed that PSMA PET-CT had pooled sensitivity and specificity of 0.97 and 0.66, respectively, and a negative likelihood ratio of 0.05 for the initial detection of PCa in patients with clinical suspicion, using histopathology as the reference standard [33]. The low negative likelihood ratio in this analysis suggests that PSMA PET-CT may be used to rule out disease in patients with a clinical suspicion of PCa, potentially avoiding unnecessary biopsies [33]. A retrospective analysis comparing PSMA PET-CT to mpMRI investigated 144 patients who underwent both imaging modalities, finding that both had excellent rates of detecting PCa but a higher sensitivity for clinically significant PCa through PSMA PET-CT when compared to mpMRI (94.85% vs. 86.03%; *p* = 0.022) [34]. A similar but smaller retrospective study also found PSMA PET-CT to be more accurate than mpMRI in the diagnosis of PCa, whilst also showing higher rates of detection of localised bilateral disease and multifocal disease [35]. Further studies have also shown the added benefit PSMA PET-CT can have in intermediate risk PCa (ISUP grade 2 and 3) demonstrating its higher sensitivity, specificity, PPV and Negative Predictive Value (NPV) than mpMRI alone [36]. One potential advantage of PSMA PET-CT over mpMRI is that interpretation is not influenced by biopsy-related artefacts such as haemorrhage or inflammation. Another advantage is clarification of indeterminate lesions within large, hyperplastic transitional zones, which might be easily missed on biopsy.

A combination of MRI and PSMA PET-CT is also being used to improve the detection of localised PCa. The PRIMARY prospective clinical trial assessed 291 patients with suspected prostate cancer using MRI, PSMA-PET-CT or MRI + PSMA PET-CT. Included patients were MRI and biopsy-naïve men with a raised PSA or suspicious DRE [37]. Patients underwent PSMA PET-CT and MRI, and each were analysed separately. Patients proceeded to systematic transperineal prostate biopsies, with recommended minimum 18 cores and additional targeted biopsies when possible. Study found 67% of patients had a positive MRI (PIRADS 3–5), 73% had positive PSMA PET-CT and 81% patients were positive for clinically significant PCa when both modalities were used. They found the NPV for these in clinically significant PCa were 72, 80 and 91% respectively. Similarly, the false negative rate improved from 17% for MRI compared to 10% using PSMA PET-CT and just 3% using a combination of MRI and PSMA PET-CT. Of the five patient’s combined MRI and PSMA PET-CT missed, four had ISUP 2 malignancies and one had ISUP 3 malignancy [37]. In select patients, there appears to be benefits of combining the improved specificity offered by PET imaging with the improved tumour localization offered by MRI.

## 5. Local Staging

Accurate assessment of T-stage is crucial to provision of the most appropriate treatment, thereby improving BCR-free survival. A clear understanding of the relationship between suspected cancer and key local structures (seminal vesicles, neurovascular bundle, prostate apex, bladder neck and rectum) is paramount for surgical and IMRT planning. Traditionally, this is detail gleaned on MRI using direct qualitative signs, such as irregular bulging or disruption of the prostate capsule [38]. However, identification of more subtle signs rely on subjective assessment of neurovascular symmetry and focal low signal intensity in SVs or peri-prostatic fat. MRI has a known low sensitivity for focal (or microscopic) EPE [39]. A recent study further comparing PSMA PET-CT to MRI for identification of Extracapsular extension (ECE) and seminal vesical invasion (SVI) found that PSMA PET-CT had a higher sensitivity for detection for ECE compared to MRI (78 vs. 54%) but no significant difference in SVI (75 vs. 67%) [40]. Another study by Skawran et al. (2022) found that there was similar sensitivities (58% vs. 61%) and specificities (81% vs. 81%) between the two modalities [41]. When slight modifications in dissection can have such profound functional and oncological implications, it is not unreasonable that more certainty provided by PSMA PET-CT is warranted in equivocal cases where nerve sparing surgery is planned.

## 6. Conventional Staging

The advent of PSMA PET-CT has highlighted the limitations of CT and general nuclear medicine bone scans (BSs) in staging of PCa. When comparing PSMA PET-CT to conventional CT and BS, Hofman et al., (2020) reported, in a randomised controlled trial, that PSMA PET-CT was superior to CT and BS in detecting LNM (92% vs. 65%), sensitivity (85% vs. 38%) and specificity (98% vs. 91%). Furthermore, there were more equivocal lesions with CT and BS than with PSMA PET-CT (23% vs. 7%) and a higher radiation exposure in CT and BS compared to PSMA PET-CT (19.2 mSv vs. 8,4 mSv) [42]. The recent meta-analysis by Wang et al., (2021), compared the detection of LNM using multiparametric prostatic MRI (mpMRI) and PSMA PET-CT, showing PSMA PET-CT to have superior sensitivity (71% vs. 40%) and similar specificity (92% vs. 92%) (46). They also demonstrated that PSMA PET-CT was able to detect smaller lymph nodes with an average avid diameter of 7 mm compared to 11.3 mm using MRI [43]. Several studies have compared the predictive ability of PSMA PET-CT with their histopathological outcomes. Across these studies the sensitivities had a wide range on a per lesion analysis, varying from 33–92%, and specificities ranging from 82–100% (Figure 2) [44,45,46,47,48].

One of the limiting factors for the utility and accuracy of PSMA PET-CT has been the variability and size of LNM. In the study by Budaus et al., (2016), the median size of detected vs. undetected LNM using PSMA PET-CT differed by a mean of 9 mm resulting in 67% of patients in their study, identified as negative for LNM based on PSMA PET-CT, ultimately returning as positive on final histopathology [44]. Similarly, Van Leeuwen et al., (2017), found that the mean diameter of true positive lymph nodes were around 5 mm larger than those with false negative LNM and concluded that almost all (91%) of LNM <5 mm in diameter and all LNM <2 mm in diameter were undetectable by PSMA PET-CT [44,48]. Furthermore, Van Kalmthout et al., (2020) performed a prospective multicentre trial on patients with newly diagnosed PCa with negative Bone scans and MSKCC >10% of nodal involvement. In their study 97 patients underwent ePLND for increased risk of LNI and found that PSMA PET-CT had a 42% sensitivity and 91% specificity [49]. This highlights that PSMA PET-CT is an improvement in imaging for the detection of LNM but not a perfect imaging modality and is not reliable for LNM <5 mm in size.

Despite this, PSMA PET-CT represents the most accurate means of staging prostate cancer and is impacting our management of PCa. The proPSMA trial, a multicentre randomised two-armed clinical trial of 302 patients, directly compared conventional (CT plus Bone scan) staging to PSMA PET-CT as first line imaging in patients with high risk PCa (PSA > 20 ng/mL, ISUP3-5 or/and clinical stage T3 disease) [42]. The study aimed to determine if PSMA PET-CT had superior accuracy than conventional imaging for the identification of pelvic LN or distant metastases, using a predefined reference standard consisting of histopathology, imaging and biochemistry at a 6 month follow up. The study found that PSMA PET-CT had 27% greater accuracy than conventional imaging (91% vs. 65%), with conventional imaging demonstrating lower sensitivity (38% vs. 85%) and specificity (91% vs. 98%). The superior accuracy of PSMA PET-CT was also demonstrated in patient subsets with pelvic nodal disease and distant metastases. Importantly, PSMA PET-CT altered management (28% vs. 15%) and had lower rates of equivocal findings (7% vs. 23%). This study also added cross-over second line imaging, which showed a management change in 27% of patients who had PSMA PET-CT as second-line, compared to 5% of patients who had conventional imaging as second line. These changes entailed a change of surgical technique (7.4%), usually the addition of an ePLND, and change in radiotherapy dose (7.4%), mostly an increase in the amount and field of radiotherapy [42]. Karagiannis et al., (2022), also found that PSMA PET-CT modified radiotherapy treatment plans in approximately 60% of their patients, usually finding further locoregional or metastatic disease and thus implementing additional systemic therapy [50].

## 7. ePLND in the Era of PSMA PET-CT

It is not clear, however, if ePLND should still be based on a nomogram calculation of an individual’s risk of LNM or PSMA PET-CT findings alone. Traditionally, PLND is considered only valuable for prognosis as correctly identified pN1 patients typically benefit from adjuvant therapies [51]. This is supported by research that has failed to demonstrate an oncological benefit for lymphadenectomy or its extent [52,53].With the new lens of PSMA PET-CT, new studies have demonstrated 50% of suspected LNM lie outside the boundary of ePLND [54]. While not vogue, an ePLND in appropriately selected patients may remove all positive nodes, thereby maximising local disease control.

Despite the higher accuracy of PSMA PET-CT, it seems clinical factors should still guide decision to perform ePLND. A recent meta-analysis demonstrates that the sensitivity of nodal staging does not exceed 60% [55]. PSMA radio-guided surgery is also being investigated and has demonstrated some success in intra-operative identification of LNM that are too small for the spatial resolution of PSMA PET-CT or MRI [56]. The EAU currently recommends the use of novel nomogram developed by Gandaglia et al. [57], which includes mpMRI targeted biopsy results as one of its parameters, along with PSA, clinical stage and maximum diameter of index lesion on mpMRI. This nomogram had an AUC of 86%, compared to 82% for Briganti 2017 nomogram. Utilising a cut-off of 7% to identify candidates for ePLND, this nomogram missed 1.6% of patients with LNM (compared to 4.6% if using Briganti 2017 nomogram). Interestingly, Meijer et al., (2021) assessed the predictive performance of the commonly used Briganti and MSKCC nomograms with the addition of PSMA PET-CT and found a substantial improvement in discriminative ability, from AUCs of 0.76 to 0.82 for Briganti 2019, and 0.71 to 0.77 for MSKCC [58].

## 8. Metastatic PCa and PSMA PET-CT

European Association of Urology guidelines currently recommend the use of PSMA PET-CT for assessment of metastases [59]. In population-based studies PCa most commonly spreads to bone (84%), distant lymph nodes (10.6%), liver (10.2%), and thorax (9.1%) [60]. Approximately 18% of men have multiple metastatic sites involved (Figure 3). About 10% of patients with PCa have bone metastasis at presentation, and 33% of the remaining patients will develop metastases during follow-up [61]. Traditional imaging modalities may be helpful in evaluating distant metastases, with CT able to detect sclerotic bone lesions and visceral metastases, however CT has been reported to be positive in only 14% of cases [62]. BS has been up until recently the most widely used method to detect bone metastases in clinical practice due to its low cost. It can detect bone metastases with good sensitivity and can carry out whole body skeletal examination, however it is non-specific, with inflammation confounding metastatic deposits. Moreover, it reportedly only has a positive rate of 5% when PSA < 7 ng/mL [62], making it an imaging technique more suited to patients with very high PSA ranges and suspected late-stage disease.

A recent meta-analysis [63] aimed to compare the detection of bone metastases of PCa between PSMA PET-CT, NaF-PET-CT, choline-PET-CT, MRI and BS. They demonstrated PSMA and NaF-PET had higher pooled sensitivities on a per-patient basis (0.97 and 0.96 respectively) than choline-PET, MRI and BS (0.87, 0.91, 0.86). Further prospective trails have showed similar results with greater sensitivity (96% vs. 73%) and better specificity (99% vs. 84%) in the detection of skeletal metastases [64,65]. This improvement has profound implications in the management of small volume metastatic disease invisible on conventional staging.

Liver metastases typically occur in systemic, late stage disease, however there are cases of patients with liver metastases as the only metastatic site, thus early detection remains important for treatment decisions. There is evidence that PCa metastases to the liver are associated with neuroendocrine characteristic [66], and this malignant pattern might lead to the loss of PSMA-expression [67], which would hamper the visualisation of liver metastases by PSMA PET-CT. In a retrospective study, Damjanovic et al. (2019) reviewed 739 PCa patients for hepatic metastases using PSMA PET-CT together with CT or MRI. A total of 17 patients had hepatic metastases, with 15 patients (83.3%) demonstrating PSMA-positive metastases, two patients (11.1%) PSMA-negative metastases, and one patient (5.6%) had mixed metastases [68]. This study was limited by lack of histopathological confirmation of results as no liver biopsies were performed, however it shows that while PSMA PET-CT remains robust at 83.3% detection of PCa liver metastases, it’s limitation lies in the reliance of cellular expression of PSMA, which can be lost with disease progression and tumour dedifferentiation.

Pulmonary metastases are considered the second most common extra-nodal metastatic site for PCa in autopsy studies (lung 46% vs. bone 90%) [69], and its reliable detection as PCa, as opposed to a concurrent primary lung malignancy or benign process, is of high clinical importance for staging and management. Retrospective studies have found PSMA PET-CT to detect 72.5% of pulmonary metastases [70]. The PSMA PET-CT negative lesions (27.5%) were postulated to be secondary to loss of PSMA due to neuroendocrine transdifferentiation (confirmed histologically in a single case where the metastatic deposit was biopsied [70]), however Pryka et al., (2016) demonstrated that due to high PSMA uptake in lung cancer, PSMA PET-CT was unable to differentiate between a lung primary lesion and PCa metastasis [71]. The utility of PSMA PET-CT in assessment of lung lesions might be further restricted, as benign lesions such as areas of bronchiectasis [72], sarcoidosis [73] and tuberculosis [71].

Despite PSMA PET-CT improving the detection rate in early recurrence, there are clinical challenges to its use, primarily due to technical shortcomings including a short half-life (^68^Ga has a physical half-life of only 68 min [74]) and limited availability of ^68^Ga. It is known from PSMA PET-CT studies with different ligands that PCa lesions are shown with better contrast and higher tracer uptake after longer uptake times (eg. 3 h, rather than 1 h after injection which is the standard protocol) [75,76,77]. Thus, imaging with a ligand with longer half-life and higher activity (such as ^18^F(Fluorine)-PSMA-provides for higher lesion uptake and superior clearance of background activity. Interestingly, ^18^F-PSMA-1007 has also been found to have less urinary activity than PSMA PET-CT, which would improve its differentiation of local recurrence and regional lymph node metastases from ureter/bladder activity, and decrease rate of false positives [76,78].

Furthermore, as mentioned prior, PSMA is not exclusively expressed in PCa. This uptake in other parts of the body can potentially increase the difficulty of interpretation of PSMA PET-CT in suspected metastatic disease. A comprehensive prospective trial by Fendler et al., (2021) found that in patients post radiotherapy or RP who met criteria for BCR, (PSA > 0.2 mg/mL post RP or PSA > 2 ng/mL above nadir following radiation therapy) 17 of 217 patients (8%) had a false positive PSMA PET-CT Of these, almost two-thirds occurred in the context of suspected recurrence in the prostate post radiotherapy. Other causes for false positives included one case of primary lung cancer, one bronchogenic cyst, one prostatic abscess and two cases of fibrosis [79].

Another important consideration is the small proportion of reportedly negative PSMA PET-CT in the context of raised PSA [80] (e.g., PSA > 10 ng/mL, where negative PSMA PET-CT was 4% [81]) and metastatic hepatic and pulmonary lesions which are PSMA PET-CT negative. According to literature almost all prostatic adenocarcinomas will express PSMA [82], however there is a subpopulation that lacks strong PSMA tracer uptake, including men with neuroendocrine histology. Those men with advanced, castration resistant disease, may have areas of de-differentiation and loss of PSMA expression [25]. False negatives are also more common in patients with lower serum PSA values, or slower PSA kinetics. For purely intraductal carcinoma, which represents around 0.3% of all prostate cancers [83], the sensitivity of PSMA PET-CT has been questioned. Intraductal PCa has been shown to have a lower PSA expression by 30% and thus may make detecting intraductal PCa more difficult [84]. No specific studies have reviewed the efficacy of PSMA PET-CT in intraductal PCa, however several articles express concerns over their accuracy and suggest the addition of mpMRI or FDG PET to more accurately stage and monitor patients [85,86].

## 9. Stage Shift & Evolution of Oligometastatic Prostate Cancer

The current treatment paradigm for patients with rising PSA after maximal local therapies with negative conventional imaging is non-curative, consisting of systemic treatments. Amongst others, this algorithm is based on the results of the CHAARTED and LATITUDE trials, in which conventional imaging was used to detect metastatic disease [87,88,89]. In other words, patients with molecular PSMA-identified only oligo-recurrent or de novo synchronous oligometastatic disease were not included in these studies. Therefore, the recommendations of these seminal papers need to be interpreted carefully in patients with positive PSMA PET-CT but negative conventional imaging.

Similarly, many patients in previous literature considered to have high-risk localised disease were probably oligometastatic. While there is limited data in this space, we know that men with de novo oligometastatic disease in the H-arm of the STAMPEDE trial derived a 10% 3-year OS benefit from local radical radiotherapy in addition to systemic treatment, compared to those who received systemic treatment alone [90]. It is not unreasonable to extrapolate and expect similar outcomes for cytoreductive prostatectomy in this setting.

## 10. PSMA PET-CT and MDT (Metastasis Directed Therapy)

Metastatic PCa is becoming more accurately diagnosed and detected earlier through imaging such as PSMA PET-CT. Metastatic directed therapy is a newer concept aiming at improving outcomes for patients with oligometastases or metastatic disease. Historically, metastatic PCa was managed with chemotherapy, androgen biosynthesis inhibition, androgen receptor inhibition or radium 223 [91]. However, several techniques have been established, specifically targeting metastases. Salvage ePLND has been shown to delay the development of a new clinical recurrence [92,93]. Yet, in studies with longer term follow >5 years, the efficacy and reduction in BCR is not as promising as once thought and therefore, salvage PLND should be perceived as a technique to delay BCR rather than a cure [94]. Essentially, salvage PLND is a form of metastectomy and cure for PCa is unlikely to be achieved with patients likely requiring salvage ADT and or chemotherapy and ultimately progressing towards CRPC.

Stereotactic Body Radiotherapy (SBRT) is another metastatic directed therapy which has been enhanced through the use of PSMA PET-CT. Several authors have shown higher disease free survival rates (64% vs. 34%), and lower long term requirement of ADT administration when using ^68^GA- PSMA PET vs. ^18^F-Choline for directed oligometastatic PCa treatment [95,96]. Similar benefits of using PSMA PET-CT to target skeletal oligometastatic disease have been demonstrated with over 40% of patients showing no evidence of disease progression [97]. Furthermore, elective nodal radiotherapy has also been shown to have a potential benefit for survival and decrease BCR in a recent systematic review. De Meerleer et al., (2021) found that patients with high risk PCa and evidence of pathologically positive pelvic lymph nodes predominately diagnosed through PSMA PET-CT had a substantial benefit with elective nodal radiotherapy, with minimal grade III or higher toxic effects [98].

## 11. Role of PSMA PET-CT in Biochemical Recurrence after Curative Treatment

The recurrence of PCa is defined by a rise in serum PSA level, termed biochemical recurrence (BCR), which occurs in 20–30% of patients after radical prostatectomy and up to 60% of patients after radiotherapy [99]. On average BCR precedes the appearance of clinical metastasis by 8 years [99]. The definition of BCR is a serum PSA over 0.2 ng/mL in two separate tests after prostatectomy, or an absolute rise in PSA level of 2 ng/mL over the posttreatment PSA nadir following radiotherapy (ASTRO-Phoenix consensus definition) [100]. Apart from a rise in PSA, it is difficult to detect early recurrence as symptoms are absent with low disease burden. Treatment strategies for PCa recurrence varies according to the presence of local recurrence, loco-regional lymph node involvement, or metastases to distant organs or bone [101], with treatment options including local salvage therapy, systemic therapy, surveillance or androgen deprivation. Thus, it is important for patients with BCR to have early and correct identification of the extent of recurrent disease to guide management decisions.

Previously, guidelines have recommend traditional imaging modalities such as CT, BS and MRI in the setting of BCR, however they are limited in their ability to detect small lesions, with limited sensitivity at lower PSA levels (PSA < 2 ng/mL) [59]. CT has poor anatomical resolution in the treated prostate bed, and unless recurrence is large, it is of limited use. One meta-analysis found CT sensitivity was 0.42 (0.26–0.56 95% CI) and specificity was 0.82 (0.8–0.83 95% CI), similarly for MRI, the sensitivity was 0.39 (0.22–0.56 95% CI) and specificity was 0.82 (0.79–0.83 95% CI) [102]. Indeed, anatomical imaging techniques depend solely on morphological features for identifying recurrent disease, which is insufficient when 80% of nodal metastases from PCa are in non-enlarged lymph nodes [103]. BS has proven to be inadequate for early BCR assessment, with 4% of scans positive with PSA < 10 ng/mL [104]. FDG PET-CT has low sensitivity for BCR, with only 28% detection when PSA is <1.5 ng/mL. ^18^F- or 11C-choline PET-CT is only of useful when PSA is >2.0 ng/mL, and 11C-choline PET demonstrating only 21% detection rate when PSA < 0.4 ng/mL. Comparison between 11-C-choline, ^18^F-fluciclovine and Ga-PMA PET-CT in patients with BCR, have demonstrated that PSMA PET-CT has a superior detection rate [105].

PSMA PET-CT has been found to outperform traditional imaging methods in its detection of recurrent disease in several retrospective studies and meta-analyses, In the largest retrospective (unblinded) study by Eiber et al. (2015) 248 patients were identified and reviewed with BCR. They found that the detection rate of recurrence using PSMA PET-CT was 89.5% with a median PSA 1.99 ng/mL, and 57.9% when the PSA was <0.5 ng/mL [106]. Similarly, a study by Afshar-Oromieh et al. (2015, 319 patients) looked at BCR for patients post prostatectomy, radiotherapy, ADT or a combination of these. They found a detection rate for BCR with PSA 1.1–2.0 ng/mL was 71%, PSA of 0.21 to <0.5 was 50%, and PSA < 0.2 ng/mL was 47.1% as well as a patient-based sensitivity of 88.1% [75]. Fendler et al., (2019) performed a prospective multicentre trial of 635 men with BCR-PCa, found a direct correlation between detection rate as serum PSA increased: 38% for <0.5 ng/mL, 57% for 0.5 to <1.0, 84% for 1.0 to <2.0 ng/mL, 86% for 2.0 to <5.0 ng/mL, and 97% for ≥5.0 ng/mL [107]. The overall detection rate was 75%. Their results found an overall positive predictive value for ^68^Ga-PSMA-11 PET of 0.92 (95% CI, 0.88–0.95) [75]. A proportion of these patients underwent histopathological correlation and, with similar results to Afshar-Oromieh et al. (2015), the PSMA PET-CT positive predictive value for localisation of PCa was >0.8 for a median PSA of 2.1 ng/mL [75].

To date there have been multiple meta-analyses assessing the utility of PSMA PET-CT [108,109,110,111,112,113] to detect recurrence. In patients with BCR, the proportion of positive PSMA PET-CT scans consistently increased with higher PSA. For PSA categories 0–0.19, 0.2–0.49, 0.5–0.99, 1–1.99, and >2 ng/mL, the percentages of positive scans were 33%, 45%, 59%, 75%, and 95%, respectively. Significantly, PSMA PET-CT improves the detection of recurrence at low PSA (<0.5 ng/mL), and has high sensitivity (75%) and specificity (99%) on meta-analysis of pooled data (Figure 4) [108]. Furthermore, a meta-analysis by Kimura et al., (2020) looking at accuracy for PSMA PET-CT performed prior to salvage lymph node dissection for nodal recurrence in PCa patients with BCR (462 patients, 14 studies) found sensitivity using lesion based analysis of 0.84 (95%CI: 0.61–0.95) and specificity 0.97 (95% CI: 0.95–0.99) [113].

Shortcomings notwithstanding, uptake of PSMA PET-CT by clinicians has been enthusiastic, with detectable impact on patient management. A prospective study demonstrated the change in therapeutic planning in 62% of cases with BCR after definitive management, with PSMA PET-CT detecting unsuspected: local relapse in 27% patients, lymph node spread 39%, distant metastases 16% [114]. A retrospective single centre study, found PSMA PET-CT changed staging in 32% of patients, with 23% upstaged, and 9% downstaged after PSMA PET-CT (Figure 5) [115], compared to conventional imaging, leading to changes in treatment strategies. Meta-analysis results are more conservative, with 54% of patients reporting change in intended management after PSMA PET-CT [116], with changes including proportionally higher rates of radiotherapy (56% to 61%), surgery (1% to 7%), focal therapy (1% to 2%) and multimodal treatment (2% to 6%), and lower rates of systemic therapy (26% to 12%) and no treatment (14% to 11%) after PSMA PET-CT in patients with BCR.

PSMA PET-CT for BCR has identified differences in sites of recurrence when comparing BCR post radiotherapy and prostatectomy. Lawal et al. (2021) identified 247 patients with BCR post external beam radiotherapy (EBRT) or RP. When comparing EBRT and RP, the site of recurrence on PSMA PET-CT was 67.4% vs. 43.4% at the prostate bed, 17.4% vs. 16.5% at SV, 37.2% vs. 34.7% in pelvic nodes and 9.3% vs. 18.3% to bone, respectively, [117]. Similar results were demonstrated in a review by Armstrong et al. (2020), which also found that patients who underwent RP had a lower local recurrence rate than those who had radiotherapy (21.3% vs. 63.3%) but shorter time from initial therapy to recurrence (27.7 months vs. 54.6 months). They also found that recurrence rates were lower in RP patients compared to those that received radiotherapy for extrapelvic metastasis (20.0% vs. 46.8%) [118].

In patients with BCR, PSMA PET-CT is currently being used as the gold standard imaging modality to identify residual cancer and guide planning of salvage radiotherapy (SRT) planning [119]. Interestingly, retrospective studies using PSMA PET-CT of patients with biochemical persistence (BCP) post-radical prostatectomy have found the obturator, mesorectal and pre-sacral nodes at highest risk of containing residual disease [120]. In this cohort of patients, SRT represents one of mainstays of treatment although the role of concurrent ADT is still being investigated. A retrospective study by Rogowski et al. (2022), demonstrated that PSMA PET-CT based SRT for lymph node recurrence yielded BCR-free survival of 81%, 72% and 66% at 1, 2 and 3 years, respectively, [121].

## 12. Limitations of PSMA PET-CT

Despite it being a non-invasive, readily performed, swift process there are some limitations and side effects of PET and whole-body PET-CT. PET does require exposing patients to radiation, however, this dose is relatively low at approximately 2.0 mSv for PSMA PET-CT alone and an additional 22 mSv if whole body PET-CT was performed totaling approximately 24 mSv [107]. Serial radiation exposure through CT has been shown to have and estimated lifetime attributable cancer mortality risk of approximately 2% with annual exposure for 30 consecutive years [122]. Most patients would receive one or two PET-CT’s in their lifetime thus making the cancer mortality risk negligible. Interpretation of PET imaging may be difficult in patients who are unable to lie still for their scan, potentially requiring reimaging or sedation to improve image quality as well as potential for interdepartmental and international variation in protocol timing of tracer administration and SUV measurement [123]. Furthermore, reliance on the production of radiotracer allows PSMA PET-CT to be susceptible to broader changes in the global community with supply chains dependent on a cohesive global community.

From a clinical perspective, PSMA PET-CT has been well documented to detect PCa with positive scans seen in the majority of patients with suspected cancer (approximately 83%) and shown to be highly specific. Despite superior accuracy compared to cross-sectional imaging, false negatives have also been described including small nodal metastases under the special resolution of PET (i.e., <5 mm), ISUP < 3 tumours and a minority of prostate cancers [124]. Mannweiler et al. found that 5% of primary prostate cancer and 15% of prostate cancer mets were PSMA-negative in immunohistochemistry [125].

Furthermore, stage migration based on the accuracy of PSMA PET-CT has also become an area of interest with patients being upstaged now representing more favourable disease compared to others in the new stage grouping—apparent rates of survival improve, yet there is no impact on individual patient outcome (also known as the Will Rogers phenomenon).

## 13. PSMA PET-CT in the Future

Moving forward, it is likely that PSMA PET-CT will provide even more accurate diagnosis and staging for PCa. This could occur through the development of new radioligands. ^18^F-PSMA-1007 is one of the newly developed radioligands that may improve the accuracy of diagnosis and staging. It has different properties to PSMA PET-CT including a longer half-life and reduced urinary clearance [126]. Initial studies show benefit at identifying small LNM (≥3 mm) with a sensitivity of 82% and specificity of 99.6% [127]. However, the rate of detection of benign lesions in thyroid, non-specific lymph nodes and bones requires careful interpretation of the PET images to avoid false positive results [128].

Furthermore, radiotheranostics and the use of radioligands with specific uptake targeting PSMA have been proposed to treat PCa. These new radioligands have been recognised to deliver a sustained treatment to the local microenvironment without causing harm to surrounding tissue [129,130]. As mentioned earlier, PSMA is not exclusively expressed on prostatic tissue and can be identified in other malignant and non-malignant processes throughout the body. Lutetium-177-PSMA-617 (^177^LuPSMA) is a radioligand which delivers beta particle radiation selectively to PSMA positive cells and their direct microenvironment. The VISION trial followed 831 patients with castrate resistant prostate cancer and randomly assigned them to the ^177^LuPSMA arm or the control group. They showed an increase in overall survival from 11.3 to 15.3 months (*p* < 0.001), decrease in PSA level by 50–80% and delayed imaging-based progression through CT, MRI or BS. This treatment did have significant side effects with 53% of patients suffering grade 3 or higher adverse events compared to 38% of the control arm (134). Further PSMA ligands using copper (^64^Cu) and scandium (^44^SC) have also been proposed with different radioligand properties such as significant tumour uptake and retention after 24 h, appearing to be promising alternatives [131]. Yet, all of these are still in their infancy but provide a glimpse of the future for PSMA PET-CT and PCa.

Another area PSMA may prove to be useful is through the tracking of radioactively tagged LNM in intraoperative PLND. After performing PSMA PET-CT and defining patients with high risk prostate cancer suspicious for positive pelvic lymph nodes, a radiotracer is injected prior to surgery that emit specific fluorescence or gamma-emitting tracers such as ^99m^Tc or ^111^InPSMA, which are then able to be detected through direct vision or gamma-emitting probe detectors [132,133]. Although in its infancy, real time diagnosis and treatment of positive lymph nodes accessible during operative management has the potential to decrease BCR and increase overall survival. In addition to this, xenograft models have been used to assess the utility of alpha particle emitting analogues of ^213^Bi-L1 and ^225^Ac-L1. Several studies suggest comprehensive cell uptake on a micrometastatic level and therapeutic effect on PSMA positive cells with minimal toxicity. [134,135].

In summary, PSMA PET-CT is now routinely used in the evaluation of prostate cancer in the context of primary staging and suspected tumour recurrence. More specific PET radiopharmaceuticals my improve both assessments and provide additional benefits in theranostics. PET-MR with the ability to perform simultaneous PET and MR acquisitions may be the imaging modality of choice in the future.

## Figures and Tables

**Figure 1 diagnostics-12-02594-f001:**
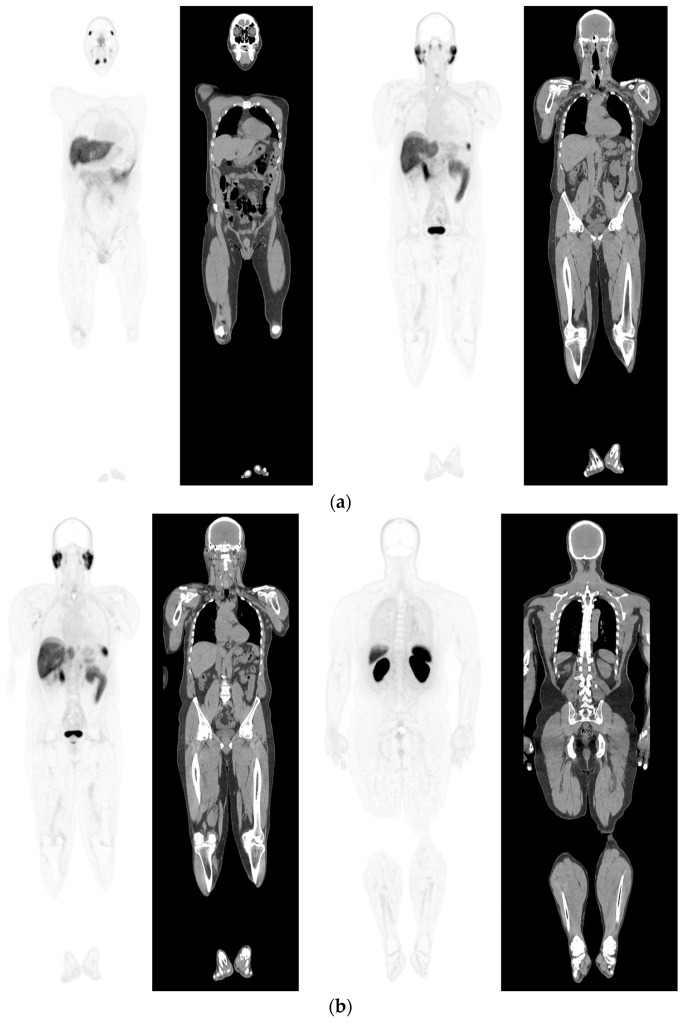
PSMA PET-CT scan showing normal distribution of PSMA-11. 60 yr old M—Gleason 6 prostate cancer on active surveillance for past 2 years; PSA 7.2; MR imaging—low grade changes PIRADS 2. PSMA PET-CT: ^68^Ga-PSMA - 204 MBq; uptake 52 min; BMI = 29.7; Wt 90 Kg; coronal PET and corresponding CT slices (soft tissue windows) from left to right. (**a**)—physiological uptake in lacrimal, submandibular salivary glands, parotid glands, retropharyngeal soft tissue, liver, bowel, part of spleen and pooling of tracer in bladder; mild reactive tracer uptake in groin and axillary nodes. (**b**)—physiological uptake in head and neck; reactive uptake axillary nodes; focal uptake in apex of prostate gland anterior below the bladder SUV = 9.1; marked uptake/excretion of tracer in both kidneys.

**Figure 2 diagnostics-12-02594-f002:**
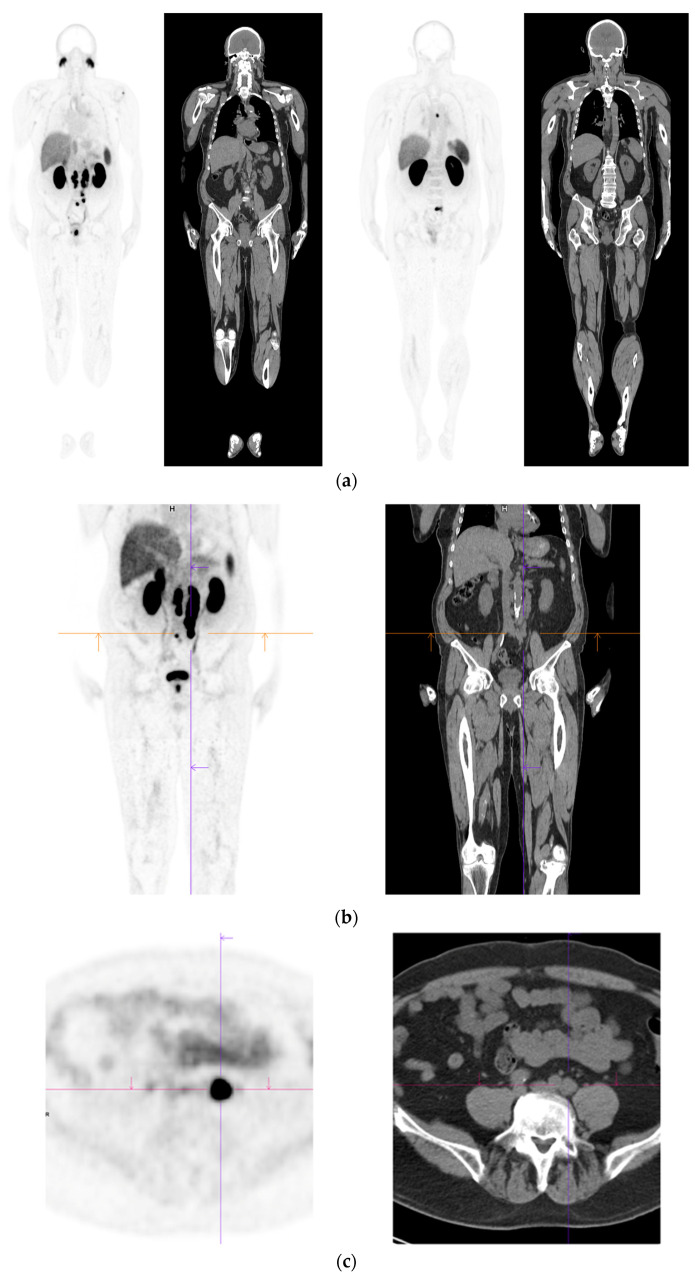
Primary staging. 70 yr old M; PSA 42; Gleason 4 + 5 = 9; MR scan—PIRADS 6 at right apex; PIRADS 4 at left midzone. PSMA PET-CT: ^68^Ga-PSMA - 216 MBq; uptake 48 min; BMI = 30.5; Wt 83 Kg. (**a**): Coronal images—focal uptake midline apex anterior SUV = 26.1 with smaller focus SUV-5.4 right midzone; bulky disease in abdominal nodes; metastasis left humerus; left para-aortic node in thorax; bony metastasis left side of S1. (**b**,**c**): enlarged coronal and transaxial images of left common iliac/para-aortic nodal disease and uptake in apex of gland; transaxial images—node measures 14 mm SUV = 40.1; X-hairs show location. (**d**,**e**): enlarged coronal and transaxial images of right pelvic nodal that measures 1.4 mm SUV = 5.1; uptake in right midzone and apex as well as disease in left midzone of gland; X-hairs show location. (**f**,**g**): enlarged coronal images of para-aortic/para-oesophageal nodal disease in mediastinum; node measures 5.5 mm, SUV = 17.2; X-hairs show location.

**Figure 3 diagnostics-12-02594-f003:**
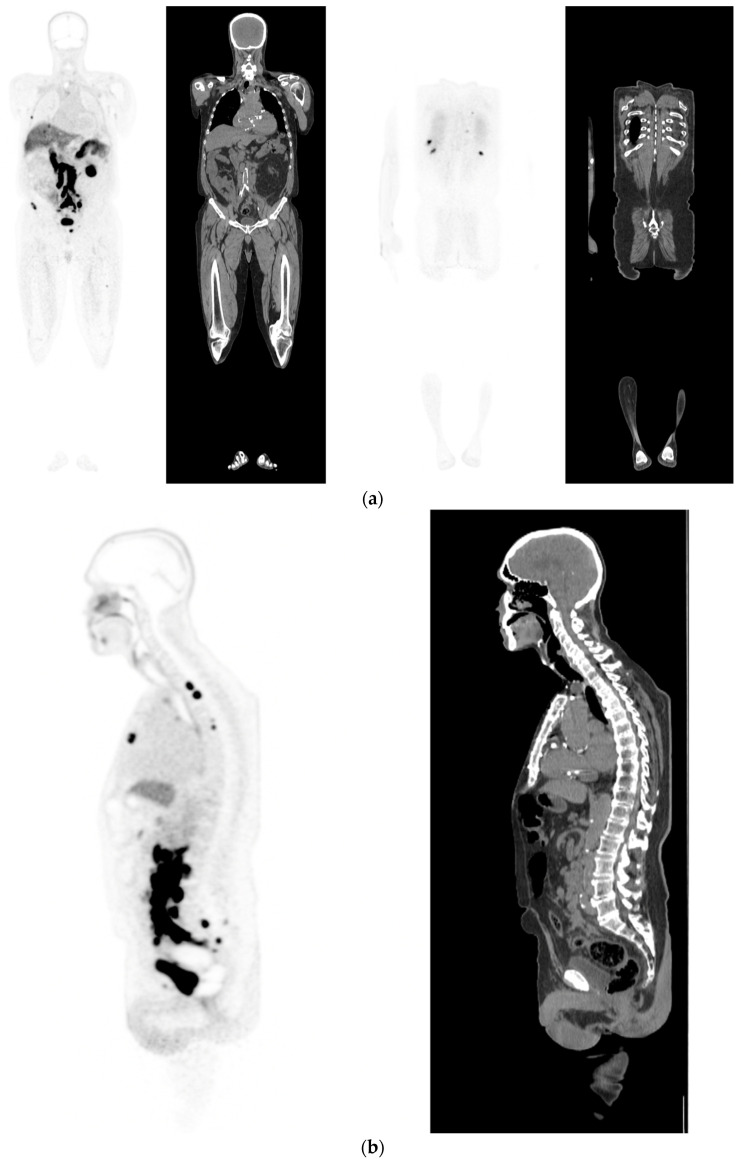
Recurrent disease. 74 yr old M; radical prostatectomy 8 yrs prior; extensive para-aortic and upper pelvic lymph node involvement on CT; severe back pain; PSA 60. PSMA PET-CT: ^68^Ga-PSMA-216 MBq; uptake 48 min; BMI = 31.9; Wt 89 kg. (**a**,**b**): Coronal and sagittal images show bulky nodal disease with markedly increased uptake (SUV = 68.2) in abdomen and pelvis, hepatic metastases, multiple bony metastases in pelvis, lower limbs, sternum, ribs and vertebral column. (**c**): enlarged transaxial images of lower thorax/upper abdomen show tracer avid foci in segments 7 (SUV = 8.8), 4 (SUV = 8.7) of liver; X-hairs on lesion in right 10th rib, SUV = 14.6; CT on bone windows shows small region of sclerosis vs. extensive uptake on PET.

**Figure 4 diagnostics-12-02594-f004:**
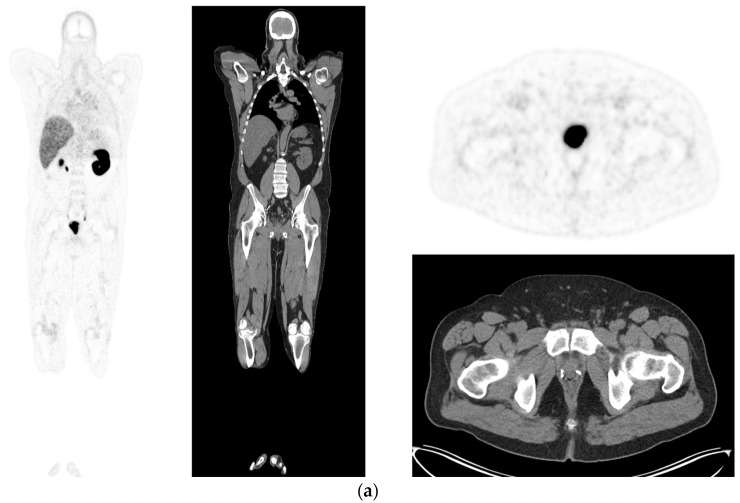
Biochemical recurrence—serial scans. 58 yr old M—radical prostatectomy and node disSection 10 yrs prior; scans done in July 2020 when PSA 0.24 and Sep 2021 when PSA 0.38. (**a**): PSMA PET-CT: July 2020-^68^Ga-PSMA-226 MBq; uptake 49 min; BMI = 35.1; Wt 105 kg; coronal images and transaxial images at level of prostate bed show tracer excretion; rest of study clear. (**b**): PSMA PET-CT: Sep 2021-^68^Ga-PSMA-238 MBq; uptake 50 min; BMI = 35.2; Wt 107 kg; coronal images and transaxial images at level of prostate bed show tracer excretion but with a new small focus of uptake SUV= 6.0 in left side of prostate bed; rest of study clear.

**Figure 5 diagnostics-12-02594-f005:**
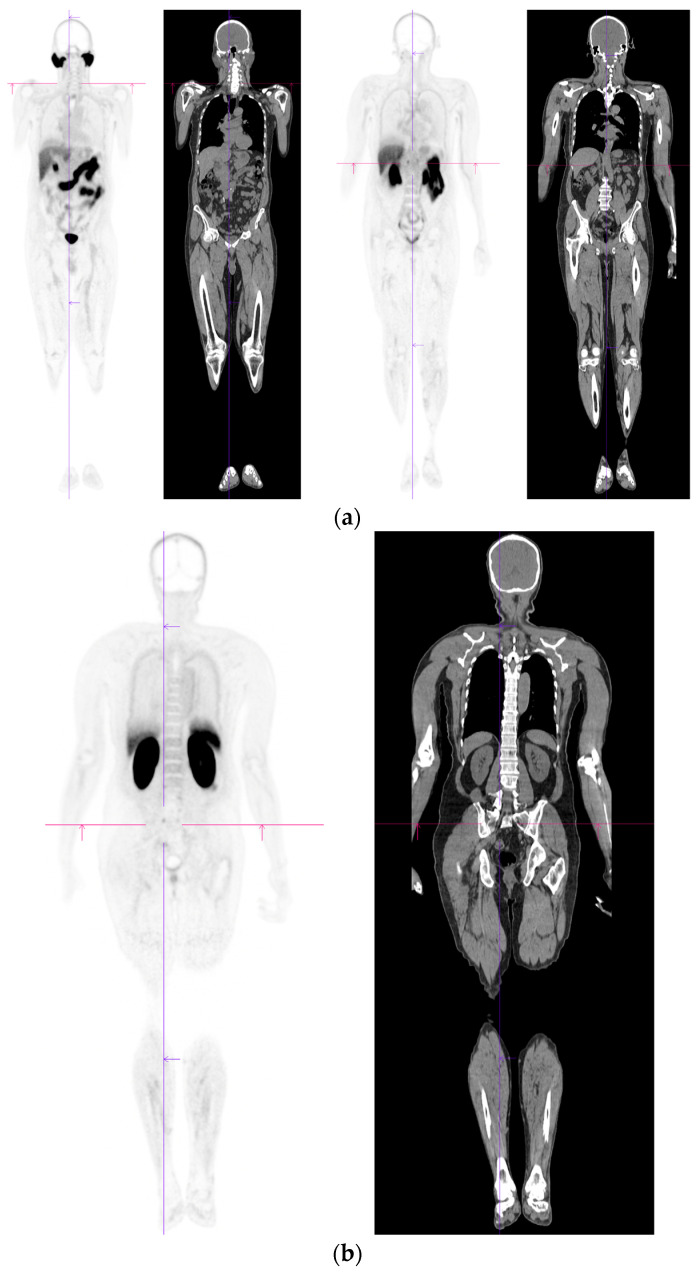
Biochemical recurrence—negative scan; normal distribution of tracer. 75 yr old M—radical prostatectomy 7 yrs prior; PSA 0.42. PSMA PET-CT: ^68^Ga-PSMA - 219 MBq; uptake 48 min; BMI = 22.6; Wt 63 kg. (**a**,**b**): coronal images show physiological uptake in parotid glands, bowel and kidneys and tracer pooling in bladder; X-hairs show uptake in cervical (SUV = 2.3) and sacral (SUV = 2.5) nerve roots and sympathetic ganglia in upper abdomen SUV = 4.6. (**c**,**d**): enlarged coronal, sagittal and transaxial images of prostate bed; X-hairs identify locations; no abnormal tracer uptake but pooling of trace in bladder problematic; Note—scale in sagittal images on (**c**) has been deliberately altered to ensure there are no adjacent focal regions of uptake to suggest local tumor recurrence.

## Data Availability

Not applicable.

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
