# Peer review of "PSMA PET-CT in the Diagnosis and Staging of Prostate Cancer"

_diagnostics, 2022, doi:10.3390/diagnostics12112594_

Round 1

Reviewer 1 Report

The paper under evaluation is a very thorough and accurate review focused on a cutting-edge topic, that is the role of PSMA-PET in PCa. I congratulate with the authors.

Images are very illustrative. The paper is very well written.

I don't have particular concerns. Only these suggestions:

1) Please check the text; sometimes the authors write PET/CT and sometimes PET-CT;

2) As concerns the future applications of PSMA in theranostics, the authors cite, aside 177Lu-PSMA, the compounds labeled with copper and scandium. I suggest to mention also PSMA-targeted alpha therapy (i.e. 225Ac/213Bi-PSMA) which has been gaining consideration in clinical practice, also citing some references (i.e. https://doi.org/10.2967/jnumed.121.263618; doi.org/10.1080/14737140.2020.1814151)

Author Response

Thank you for reviewing our paper. Your comments are very much appreciated.

Please see responses to comments below:

1. This has been addressed. In the introduction it now states 

PSMA PET - CT will refer to 68GA- PSMA PET - CT in this article unless specified otherwise. All areas with PSMA/PET have been adjusted accordingly

2. After reviewing several articles, details of the emerging alpha therapy have been included in the future of PSMA PET-CT 

Dr Alexander Combes

Reviewer 2 Report

The authors reviewed the use of PSMA PET in prostate cancer. This review is comprehensive and can help clinicians to realize the clinical value of PSMA PET.

However, some points that the authors may consider revising:

1. In the introduction, the authors wrote, "By all measures, PCa is the most common cancer in men and the second leading cause of cancer deaths amongst men [1]." The reference described the 2018 global cancer statistics, and prostate cancer was the 2nd leading cause of cancer-related death in men. However, the 2020 cancer statistics (CA Cancer J Clin 2021;71:209-249.) showed that prostate cancer became the fifth leading cause of cancer death among men. The authors may consider updating the cancer statistics for prostate cancer.

2. In section "10. PSMA-PET and MDT (metastasis directed therapy)."

The authors wrote: Several authors have shown higher disease-free survival rates (64% vs 34), and lower long-term requirement of ADT... A missing "%" in the parentheses brackets.

3. The isotope mass number style seemed inconsistent throughout the article. Some used superscripts, others did not, some with hyphens, and others did not specify the mass number. Consistency is recommended.

4. In the "12. Limitations of PSMA PET"

The authors wrote, "Despite superior accuracy compared to cross-sectional imaging, false positives have also been described including small nodal metastases under the special resolution of PET (i.e <5mm), ISUP < 3 tumours and a minority of prostate cancers  [124]." Should it be "false negatives"?

Author Response

Thank you for taking the time to read and appraise our article.

Please see notes on the reviewers comments below:

  1. This has been updated to reflect the most up to date statistics on prostate cancer mortality
  2. % has been inserted
  3. Note made at the start that 

    PSMA PET - CT will refer to 68GA- PSMA PET - CT in this article unless specified otherwise

  4. Yes - this was an error - it has been adjusted to read false negatives